# Molecular Genetic Characteristics of *FANCI*, a Proposed New Ovarian Cancer Predisposing Gene

**DOI:** 10.3390/genes14020277

**Published:** 2023-01-20

**Authors:** Caitlin T. Fierheller, Wejdan M. Alenezi, Corinne Serruya, Timothée Revil, Setor Amuzu, Karine Bedard, Deepak N. Subramanian, Eleanor Fewings, Jeffrey P. Bruce, Stephenie Prokopec, Luigi Bouchard, Diane Provencher, William D. Foulkes, Zaki El Haffaf, Anne-Marie Mes-Masson, Marc Tischkowitz, Ian G. Campbell, Trevor J. Pugh, Celia M. T. Greenwood, Jiannis Ragoussis, Patricia N. Tonin

**Affiliations:** 1Department of Human Genetics, McGill University, Montreal, QC H3A 0C7, Canada; 2Cancer Research Program, The Research Institute of the McGill University Health Centre, Montreal, QC H4A 3J1, Canada; 3Department of Medical Laboratory Technology, Taibah University, Medina 42353, Saudi Arabia; 4McGill Genome Centre, McGill University, Montreal, QC H3A 0G1, Canada; 5Laboratoire de Diagnostic Moléculaire, Centre Hospitalier de l’Université de Montréal (CHUM), Montreal, QC H2X 3E4, Canada; 6Département de Pathologie et Biologie Cellulaire, Université de Montréal, Montreal, QC H3T 1J4, Canada; 7Cancer Genetics Laboratory, Peter MacCallum Cancer Centre, Melbourne, VIC 3000, Australia; 8Department of Medical Genetics, National Institute for Health Research Cambridge Biomedical Research Centre, University of Cambridge, Cambridge CB2 1TN, UK; 9Princess Margaret Cancer Centre, University Health Network, Toronto, ON M5G 2C1, Canada; 10Department of Biochemistry and Functional Genomics, Université de Sherbrooke, Sherbrooke, QC J1K 2R1, Canada; 11Department of Medical Biology, Centres Intégrés Universitaires de Santé et de Services Sociaux du Saguenay-Lac-Saint-Jean Hôpital Universitaire de Chicoutimi, Saguenay, QC G7H 7K9, Canada; 12Centre de Recherche du Centre Hospitalier l’Université de Sherbrooke, Sherbrooke, QC J1K 2R1, Canada; 13Centre de Recherche du Centre Hospitalier de l’Université de Montréal and Institut du Cancer de Montréal, Montreal, QC H2X 0A9, Canada; 14Division of Gynecologic Oncology, Université de Montréal, Montreal, QC H3T 1J4, Canada; 15Lady Davis Institute for Medical Research, Jewish General Hospital, Montreal, QC H3T 1E2, Canada; 16Department of Medicine, McGill University, Montreal, QC H3G 2M1, Canada; 17Centre de Recherche du Centre Hospitalier de l’Université de Montréal, Montreal, QC H2X 0A9, Canada; 18Department of Medicine, Université de Montréal, Montreal, QC H3T 1J4, Canada; 19Sir Peter MacCallum Department of Oncology, University of Melbourne, Melbourne, VIC 3010, Australia; 20Gerald Bronfman Department of Oncology, McGill University, Montreal, QC H4A 3T2, Canada; 21Department of Epidemiology, Biostatistics & Occupational Health, McGill University, Montreal, QC H3A 1Y7, Canada

**Keywords:** whole exome sequencing, FANCI, ovarian cancer, cancer predisposing gene, hereditary cancer, Fanconi anemia pathway, TCGA

## Abstract

*FANCI* was recently identified as a new candidate ovarian cancer (OC)-predisposing gene from the genetic analysis of carriers of *FANCI* c.1813C>T; p.L605F in OC families. Here, we aimed to investigate the molecular genetic characteristics of *FANCI,* as they have not been described in the context of cancer. We first investigated the germline genetic landscape of two sisters with OC from the discovery *FANCI* c.1813C>T; p.L605F family (F1528) to re-affirm the plausibility of this candidate. As we did not find other conclusive candidates, we then performed a candidate gene approach to identify other candidate variants in genes involved in the FANCI protein interactome in OC families negative for pathogenic variants in *BRCA1*, *BRCA2*, *BRIP1*, *RAD51C*, *RAD51D*, and *FANCI*, which identified four candidate variants. We then investigated *FANCI* in high-grade serous ovarian carcinoma (HGSC) from *FANCI* c.1813C>T carriers and found evidence of loss of the wild-type allele in tumour DNA from some of these cases. The somatic genetic landscape of OC tumours from *FANCI* c.1813C>T carriers was investigated for mutations in selected genes, copy number alterations, and mutational signatures, which determined that the profiles of tumours from carriers were characteristic of features exhibited by HGSC cases. As other OC-predisposing genes such as *BRCA1* and *BRCA2* are known to increase the risk of other cancers including breast cancer, we investigated the carrier frequency of germline *FANCI* c.1813C>T in various cancer types and found overall more carriers among cancer cases compared to cancer-free controls (*p* = 0.007). In these different tumour types, we also identified a spectrum of somatic variants in *FANCI* that were not restricted to any specific region within the gene. Collectively, these findings expand on the characteristics described for OC cases carrying *FANCI* c.1813C>T; p.L605F and suggest the possible involvement of *FANCI* in other cancer types at the germline and/or somatic level.

## 1. Introduction

Since the first reports of pathogenic variants in *BRCA1* and *BRCA2,* the breast- and ovarian cancer-predisposing genes (CPGs), almost 30 years ago [1,2], it is increasingly evident that there are unlikely to be other major high-risk genes contributing to these cancers. For ovarian cancer (OC), pathogenic variants in *BRIP1* [3,4], *RAD51C* [5,6,7], *RAD51D* [8], and *PALB2* [9] (p. 2) have been identified but each accounts for less than 2% of sporadic OC cases [10]. Other genes, such as the mismatch DNA repair genes *MLH1* [11,12], *MSH2* [13], *MSH6* [14], and *PMS2* [15] featured in colorectal cancer (Lynch syndrome) families, have been associated with OC risk as well, although carriers are also rare, cumulatively less than 1% of sporadic OC cases [10]. Other genes have been proposed for OC risk, such as *ATM* [16,17,18,19], *BARD1* [20], [21] (p. 1), and *FANCM* [22,23], although penetrance is not yet established. Our strategy for identifying new risk genes has focused on individuals from an ancestrally defined population exhibiting genetic drift, which allowed us to identify frequently occurring variants as potential candidates [24]. In contrast to the general population, a few pathogenic variants in *BRCA1* [25,26], *BRCA2* [25,26], *PALB2* [27,28], *RAD51C* [29], and *RAD51D* [29,30] are found to frequently occur in French Canadians (FCs) in Quebec, Canada [24]. Using this approach and applying biological assays, our group reported the candidacy of *FANCI* c.1813C>T; p.L605F as a new OC-predisposing gene [31]. Heterozygous carriers of *FANCI* c.1813C>T were identified more commonly in OC families negative for *BRCA1* and *BRCA2* pathogenic variants compared to FC cancer-free controls. *FANCI* c.1813C>T cancer-free control carriers were more likely to have a first-degree relative with OC, suggesting a role in the risk of OC. In addition, this variant and other candidate variants in *FANCI* were more common in familial OC compared to sporadic OC cases from another (Australian) population. FANCI, a member of the Fanconi anemia (FA) DNA repair pathway involved in the repair of interstrand crosslinks, was shown to have abrogated function in cells expressing p.L605F. FANCI p.L605F showed reduced protein expression, was destabilized upon treatment with DNA-damaging agents (mitomycin C and formaldehyde), and exhibited sensitivity to cisplatin. Although the expression of FANCI protein was variable in OC tumours, it was shown to be highly expressed in normal fallopian tube epithelium, a purported tissue of origin for high-grade serous ovarian carcinoma (HGSC) [32,33,34,35,36,37], the most common histopathological subtype of OC. Although its role in conferring a risk on OC remains to be determined, the number of *FANCI* c.1813C>T; p.L605F carriers identified in OC cases affords an opportunity to investigate the molecular genetic features of carriers.

This study applies a bioinformatic approach to assess the genetic background in which *FANCI* carriers were identified in the context of familial OC and investigates the molecular genomic landscape of ovarian tumours from carriers using available whole-exome sequencing (WES) data. We also investigate *FANCI* variant carriers in the context of other cancer types by taking advantage of The Cancer Genome Atlas (TCGA) molecular genetic data sets. We relate our findings to current knowledge concerning moderate-to-high-risk OC-predisposing genes and other established CPGs and the biological role of FANCI.

## 2. Methods

### 2.1. Study Subjects

The study groups are described in Appendix A. Family F1528 was previously reported in a study on the histopathology of FC hereditary BC and OC families [38] and was updated with clinical data in the identification of *FANCI* as a new candidate OC-predisposing gene [31]. WES analyses were previously described [31]. Briefly, DNA extracted from peripheral blood lymphocytes was exome captured, followed by 100 bp paired-end sequencing. Variants were aligned to human genome assembly hg19 for germline variant calling.

WES data available from familial OC cases negative for pathogenic variants in *BRCA1*, *BRCA2, BRIP1, RAD51C,* and *RAD51D* (n = 13 cases (12 families)) and OC cases harbouring *FANCI* c.1813C>T; p.L605F (n = 10) have been previously described [25,26,29,31,39]. Four additional HGSC cases of *FANCI* c.1813C>T were also included, which were obtained from the Réseau de recherche sur le cancer (RRCancer) Tumour and Data biobank. All cases were self-reported FC ancestry. *FANCI* c.1813C>T-harbouring cases are also included in familial OC cases with (n = 1) or without (n = 2) *BRCA1*, *BRCA2*, *BRIP1*, *RAD51C,* or *RAD51D* pathogenic variants. There are no other cases known to be ascertained to more than one FC study group. For cancer-free FC controls, we used WES data available from CARTaGENE (n = 171) [40,41,42] (cartagene.qc.ca) and whole-genome sequencing (WGS) data from the Genetics of Glucose regulation in Gestation and Growth project (Gen3G) (n = 422) [43] to survey the germline genetic landscape variants.

WES data available from Australian HGSC cases (n = 516) [31,44] were surveyed for genetic landscape variants.

TCGA PanCancer Atlas cancer cases from the general population were investigated for germline *FANCI* c.1813C>T carriers (n = 10,389) [45] and somatic *FANCI* variants (n = 10,434) [46,47]. Age at diagnosis and sex for cancer cases is available on cbioportal.org. Cancer-free cases from the general population Genome Aggregation Database (gnomAD; gnomad.broadinstitute.org) [48] were used as a comparator for *FANCI* c.1813C>T carrier frequency.

All biological samples and associated clinical information were obtained from biobanks, where participants were recruited in accordance with ethical guidelines and approved Institutional Research Ethics Boards (Appendix A). FC OC samples were anonymized at the source by providers and were assigned unique PT identifiers to further protect anonymity. This project received approval from and was conducted in accordance with The McGill University Health Centre Research Ethics Board (MP-37-2019-4783 and 2017-2722).

### 2.2. WES Filtering and Prioritization of Variants Identified in Family F1528

Sequencing data from *FANCI* c.1813C>T carrier sisters from family F1528 were sequentially filtered (Figure 1a) for (a) rare (minor allele frequency (MAF)  ≤ 1%) variants in the general population database gnomAD [48]; (b) variants within autosomes and the X chromosome only; (c) variant allele frequency (≥20% in at least one sister); (d) variant depth (≥10 reads in at least one sister); and (e) protein-coding variants. The remaining variants were visually inspected and confirmed by Integrative Genomics Viewer (IGV) [49] and filtered, as shown in Figure 1a, where variants in both sisters had variant depth ≥10 reads, variant allele frequency ≥20% for variants called heterozygous, and variant allele frequency ≥80% for variants called homozygous. The final filter applied included a survey for rare (MAF ≤ 1%) variants in the general population from the 1000 Genomes Project [50]; the National Heart, Lung, and Blood Institute (NHLBI) Exome Sequencing Project (ESP) ESP6500SI-V2 (https://evs.gs.washington.edu/EVS/); and the Exome Aggregation Consortium [51] (ExAC). These filtering steps led to a list of variants for further annotation and prioritization. The genetic landscape variants were annotated for the type of variant effect (nonsense, frameshift, splice site, or missense) and the results from applying in silico tools that predict if the variant is located at a conserved locus; whether the variant is deleterious to the protein; or if the variant has the potential to affect splicing. These in silico tools were selected based on their best predictive performance [52]. The tools used to determine the predicted conservation of variants were Genomic Evolutionary Rate Profiling (GERP++) [53], Site-Specific Phylogenetic Analysis (SiPhy) [54], Phylogenetic P-values (PhyloP) 100-way vertebrates [55], and Phylogenetic Analysis with Space/Time Models Conservation (phastCons) v1.5 [56]. Tools used to predict the ability of the amino acid change to affect protein function (deleterious or not) were Combined Annotation Dependent Depletion (CADD) v1.4 [57], Eigen v1.1 [58], Protein Variant Effect Analyzer (PROVEAN) v1.1 [59], Meta-Logistic Regression (MetaLR) [60], Meta-Support Vector Machine (MetaSVM) [60], Rare Exome Variant Ensemble Learner (REVEL) [61], and the Variant Effect Scoring Tool (VEST) v4.0 [62]. Tools used to predict the potential of variants to affect splicing were the database of splicing consensus regions (dbscSNV)’s adaptive boosting (ADA) and random forest (RF) [63], MaxEntScan [64], and SpliceAI [65]. These tools were applied in our previous studies where the biological function of proteins aligned with in silico tool prediction [29,31].

Variants were then prioritized if they were identified in both sisters and predicted to be inherited as an autosomal dominant (heterozygous) or autosomal recessive (homozygous or compound heterozygous) trait. Variants were then prioritized if they were (a) nonsense, frameshift, or canonical splice site variants (±1–2 nucleotides away from the exon), (b) missense variants predicted to affect protein function by ≥5/7 in the in silico tools and highly conserved by ≥3/4 in the in silico tools, or (c) non-canonical splice site variants (>±2 nucleotides away from the exon) predicted to affect splicing by ≥3/4 in the in silico tools, as these variants either will not encode a protein product (nonsense-mediated decay) or could affect protein function. Variants were further prioritized if they had an MAF < 1% in cancer-free controls of FC ancestry (n = 1208 alleles) because pathogenic variants are more likely to be rare based on the rare allele hypothesis [66]. The resulting variants are henceforth referred to as genetic landscape variants. *FANCI* c.1813C>T; p.L605F met all filtering and prioritization criteria and is included in the tables as a reference but was not included in the total variant counts.

### 2.3. Investigation of Genetic Landscape Variants

Genes associated with genetic landscape variants that were identified in the *FANCI* c.1813C>T; p.L605F carriers were annotated for biological function, cellular location, encoded protein function, associated disease(s), and RNA expression in the ovaries and fallopian tubes using the Human Protein Atlas [67] (proteinatlas.org). These genes were annotated using the Cancer Hallmarks Analytics Tool [68] and a list of previously identified genes associated with hallmarks of cancer [69], which are defined as various abilities or characteristics acquired by cells in the development of cancer [70,71,72]. Genes were also characterized based on being catalogued as having any somatic variants regardless of their location in the same gene in the TCGA PanCancer OC cases [46,47] (cbioportal.org) and for their association with disease in ClinGen [73].

The genetic landscape variants were annotated for carrier or allele frequency from the available WES data of familial OC cases of FC ancestry negative for pathogenic variants in *BRCA1*, *BRCA2*, *BRIP1*, *RAD51C*, and *RAD51D* (n = 13). We also investigated this OC study group for other variants in genes where genetic landscape variants had been identified, as there may be allelic, as well as genetic, heterogeneity among CPGs. The same filtering and prioritization criteria were applied to OC cases of FC ancestry.

Genetic landscape variants were annotated for carrier frequency from the available WES data of the Australian HGSC cases. Other variants in genes where genetic landscape variants were identified were not investigated in this study group, as these samples were previously reported using a landscape approach [44].

### 2.4. Loss of Heterozygosity Analyses of FANCI c.1813C>T in OC Tumour DNA from Candidate Variant Carriers

Loss of heterozygosity analysis of *FANCI* c.1813C>T was performed by Sanger sequencing of OC tumour DNA from carriers. Extracted DNA from fresh-frozen tumours was provided by the RRCancer biobank. Previously reported primers were used [31]. Sequencing chromatograms were inspected using 4peaks (nucleobytes.com/4peaks/index.html) visualization software.

### 2.5. Somatic Genetic Landscape of FANCI c.1813C>T Carriers

Extracted DNA from fresh-frozen HGSC tumours from *FANCI* c.1813C>T carriers (n = 7) was provided by the RRCancer biobank. WES was performed at the McGill Genome Centre, as previously described [31]. Annotated Variant Call Format files were inspected for variants in genes most commonly altered somatically in HGSC: *TP53*, *BRCA1*, *BRCA2*, *RB1*, *NF1*, *FAT3*, *CSMD3*, *GABRA6*, and *CDK12* [74].

Somatic copy number alteration (CNA) profiles were generated from WES data from tumour samples and corresponding matched-normal samples using Fraction and Allele-Specific Copy Number Estimates from Tumour-Normal Sequencing (FACETS) version 0.61 [75]. Total and allele-specific read counts were extracted from tumour and normal samples based on common, polymorphic SNV loci from dbSNP version 150 [76]. The following parameters were used for copy number segmentation: a minimum total sample depth of 20, a critical value for segmentation of 350, and a minimum number of heterozygous SNPs to cluster segments of 100. Focal amplification of *CCNE1* was assessed, as it has been identified in over 20% of HGSC cases [74] and may be a therapeutic target for cyclin-dependent kinase (CDK) inhibitors [77].

DeconstructSigs version 1.8.0 [78] was used to determine the contribution of known mutational signatures associated with OC in each tumour sample. Catalogue of Somatic Mutations in Cancer (COSMIC) Single-Base Substitution (SBS) version 3.2 signatures were used as a reference (cancer.sanger.ac.uk/signatures/documents/452/COSMIC_v3.2_SBS_GRCh37.txt: accessed on 16 May 2022). Mutational signatures were compared to those associated with OC [79]. Synonymous and non-synonymous single nucleotide variants (SNVs) with at least three alternate reads were used for mutational signature analysis. The number of SNVs per sample ranged from 65 to 2560.

The somatic genetic landscape of *FANCI* c.1813C>T carrier tumours from the TCGA (n = 6) was assessed for genes most commonly altered somatically in HGSC and focal amplification of *CCNE1* using cBioPortal [46,47] (cbioportal.org).

### 2.6. FANCI c.1813C>T Germline Carrier Frequency across Different Cancer Types from TCGA PanCancer Atlas

Data from the analysis of germline pathogenic variants in TCGA PanCancer Atlas cancer cases were downloaded [45]. A Variant Call Format file was generated with all *FANCI* c.1813C>T events identified at the germline level.

The clinical and genetic characteristics, including age at diagnosis and sex, of cancer cases harbouring germline *FANCI* c.1813C>T were retrieved from cBioPortal [46,47]. These characteristics were compared to the entire TCGA PanCancer Atlas study group [45].

### 2.7. Identification of Somatic FANCI Variants in Different Cancer Types from TCGA PanCancer Atlas

All somatic *FANCI* variants were retrieved from cBioPortal [46,47] TCGA PanCancer Atlas studies. The clinical and genetic characteristics of cancer cases where somatic *FANCI* variants had been identified were also retrieved. These included the total variant count, microsatellite instability (MSI) score from microsatellite analysis for normal tumour instability (MANTIS) [80], age at cancer diagnosis, and sex. These characteristics were compared to the entire TCGA PanCancer Atlas study group.

### 2.8. Investigation of Missense Variants in FANCI Reported in Public Databases

All missense variants in *FANCI* were retrieved from ClinVar [81] in March 2022. The variants were investigated using bioinformatic criteria established for *FANCI* c.1813C>T, including those uncommon in gnomAD non-cancer controls (MAF 0.1–1%), highly conserved by ≥3/4 in the in silico tools, and predicted to affect protein function by ≥5/7 in the in silico tools. The in silico tools used were the same as those mentioned above.

### 2.9. Identification of Variants in the FANCI Protein Interactome

Familial OC cases of FC ancestry negative for pathogenic variants in *BRCA1*, *BRCA2*, *BRIP1*, *RAD51C*, *RAD51D*, and *FANCI* (n = 11) were investigated for variants in genes involved in the FANCI protein interactome. To construct a list of FANCI protein interactome genes in *Homo sapiens,* we searched STRING [82], BioGRID [83], Protein Interaction Network Online Tool (PINOT) [84], Signalling Network Open Resource (SIGNOR) [85], Molecular Interaction Database (MINT) [86], Protein Interaction Knowledgebase (PICKLE) [87], Database of Interacting Proteins (DIP) [88], and IntAct [89] (Appendix A). The literature was also searched for proteins shown to directly interact with FANCI protein experimentally by using “FANCI” as the search term (Appendix A). As germline variants in DNA repair pathway genes have already been investigated in this study group (P.N. Tonin unpublished data), we focused on genes that were part of the FANCI interactome but not involved in DNA repair pathways (n = 115). Variants were filtered and prioritized using the same criteria as described above (Figure 1a,b).

## 3. Results

### 3.1. Candidate Variants Identified in Family F1528

We reported the discovery of *FANCI* c.1813C>T; p.L605F in two siblings in family F1528 as the most plausible OC-predisposing candidate based on the association of FANCI in the Fanconi anemia homologous recombination (FA-HR) DNA repair pathway [31]. The only other variant shared between these siblings and other OC carriers of *FANCI* c.1813C>T (n = 14) is *POLG* c.2492A>G; p.Y831C, a marker found in linkage disequilibrium with carriers of *FANCI* c.1813C>T in all the populations that we studied, which remains an unlikely candidate OC-predisposing allele based on its purported function, as reported previously [31]. Using bioinformatic tools and the most recent annotation from genetic databases, we re-evaluated the WES germline data of these siblings to further investigate the genetic landscape of carrier siblings, reasoning that there may be other co-occurring potentially pathogenic variants of interest that could also be investigated in other OC families of the same FC ancestry. We only applied this strategy to this OC family (F1528), as we did not have any other examples of familial cases, especially sibling pairs affected by OC where both siblings harboured the same *FANCI* variant, which would facilitate the identification of candidate variants associated with the disease.

Using the same generated WES data used in the discovery of *FANCI* c.1813C>T, we performed a new bioinformatic analysis and applied the best-performing predictive tools to identify the germline genetic landscape variants that were shared among the siblings in family F1528 (Figure 1a). From a master list of 86,061 variants identified in both sisters, we used a filtering strategy to identify rare (MAF ≤ 1%), high-quality variants that were most likely to affect the protein-coding regions, which generated a list of 222 variants that were shared among these siblings. These variants were identified in 214 different genes and, as expected, included *POLG* c.2492A>G (Appendix A). The variants were present in the same genetic state in both siblings: heterozygous (n = 196 variants), compound heterozygous (n = 14 variants in 7 genes) or homozygous (n = 1 variant), and heterozygous X-chromosome linked (n = 11). The MAFs of these variants varied from 4 × 10^−6^ to 9.7 × 10^−3^ in gnomAD, except for 14 variants that were not found in this database. These comprised 186 missense, 14 non-canonical splice site, 9 frameshift, 6 nonsense, 3 inframe, 3 canonical splice site, and 1 stop loss variant.

To further refine the list of plausible candidates, we applied criteria to the list of 222 variants to select those of interest for further investigation, focusing on the in silico tools, type of variant, and allele frequency in FC cancer-free controls (Figure 1b). Using these criteria, we identified 18 variants with nonsense, frameshift, or canonical splice site effects. We selected 57 missense candidates predicted to be damaging based on ≥5 of 7 of our best-performing in silico tools, in keeping with the rationale that the most likely biologically relevant candidates would be predicted by the majority of the in silico tools [90]. We also selected an additional two non-canonical splice site variants that were predicted to affect splicing. As expected, this excluded *POLG* c.2492A>G as a variant of interest but not *FANCI* c.1813C>T. From this list of 76 variants, we selected 66 variants for further investigation based on their allele frequency (MAF < 1%) in population-matched FC controls. Excluding the *FANCI* variant, the 66 variants of interest were identified in 66 different genes (Appendix A).

A large proportion of the 66 variants were identified at least once in various cancer contexts, such as hallmarks of cancer [68,69] (83%), or somatically mutated in OC cases from the TCGA [46,47] (83%), as summarized in Figure 1c. Although some (9/66.14%) variants have been reported in other clinical contexts, none of the variants were found to be associated with a cancer context in ClinGen [73] (Appendix A). As shown in Appendix A, some of these genes were found in more than one group, as defined in Figure 1c. None of the 66 variants were identified in genes associated with DNA repair pathways.

### 3.2. Genetic Analyses of Variants Identified in FANCI Carrier Siblings in FC Study Groups

To further characterize our 66 variants, we reviewed the available WES data from 13 familial OC cases negative for pathogenic variants in *BRCA1*, *BRCA2*, *BRIP1*, *RAD51C*, and *RAD51D* for carrier status. These cases have been well characterized and are of FC ancestry [25,26,28,29,30,31,38,91,92,93,94,95,96,97,98]. We identified a total of four carriers from three families of variants in *PTPN22, GPD1,* and *SEC14L4* (Table 1). Notable is that none of the three *FANCI* c.1813C>T carriers from the independently ascertained cases harboured any of the 66 variants. Thus, although there may be shared FC ancestry among the carriers, none of the five independently ascertained familial OC cases in our FC study group of *FANCI* carriers harboured other potentially deleterious alleles initially identified in the index *FANCI* c.1813C>T OC cases.

Given the possibility of allelic heterogeneity, even within the FC population, as we have previously demonstrated with established OC-predisposing genes [24], we screened the same 13 familial OC cases for other plausible deleterious variants in the 66 candidate genes. Applying our bioinformatic filtering and prioritization criteria, we identified 10 different variants that were not present in our index family F1528. There were six carriers of variants in *PIWIL3*, *SCN10A*, *PCDH15*, *TEX2*, *DNAH3*, *DNAH1*, *IQCA1*, *CACNA1S*, and *MYO7A* (Table 1). Four cases were found to carry variants in two different genes: F1085-PT0134 (*CACNA1S* and *MYO7A)*; F845-PT0196 (*PCDH15* and *TEX2*); F1506-PT0136 (*PIWIL3* and *DNAH1*); and F1543-PT0137 (*DNAH1* and *IQCA1*). Both variants in F1085-PT0134 were not carried by their sibling F1085-PT0135 and thus did not segregate with the disease. Each variant was harboured by only one case; two different variants were identified in the same gene, *DNAH1*.

### 3.3. Genetic Analyses of Variants Identified in FANCI Carrier Siblings in Non-FC Study Groups

We screened our 66 variants in the available WES data from the Australian HGSC study group regardless of *FANCI* variant carrier status. We identified 70 carriers of 21 variants among 516 HGSC cases, where the majority harboured only one variant (Table 2). None of these variants were identified in any of the previously identified 10 Australian carriers of *FANCI* c.1813C>T [31]. Two carriers of other likely pathogenic *FANCI* variants (c.1264G>A; p.G422R and c.3635T>C; p.F1212S) were found to harbour variants in *ALDH16A1* and *NBAS*, respectively. Although different variants were identified in OC cases of FC ancestry in *DNAH3*, *IQCA1*, and *PCDH15*, the same variants as those found in the family F1528 sisters were identified in these genes in five Australian HGSC cases.

### 3.4. Genetic Analyses of Germline FANCI Interactome Variants Identified in FC OC Cases

Based on our previous analyses, *FANCI* c.1813C>T is the strongest candidate OC-predisposing variant identified in family F1528. Therefore, we used a candidate gene approach to investigate the germline variants in genes that encode proteins that are part of the FANCI interactome. We reviewed the available WES data from 11 familial OC cases negative for pathogenic variants in *BRCA1*, *BRCA2*, *BRIP1*, *RAD51C*, *RAD51D,* and *FANCI* to search for other candidate predisposition variants. Variants in DNA repair pathway genes were excluded, as this has been previously reported by our lab (P.N. Tonin unpublished data). We identified a total of three carriers from three families of missense variants in *EZH2*, *ANKRD55*, *MOV10*, and *LRRK2* (Table 3). Variants in *ANKRD55* and *MOV10* were identified in the same case F1506-PT0136.

### 3.5. Identification of Other Germline Potentially Deleterious Variants in FANCI

To identify additional germline potentially pathogenic variants in *FANCI,* we used ClinVar [81], which aggregates information about genomic variation and its relationship to human health. We focused our analyses on the missense variants, as the loss-of-function variants (i.e., frameshift, nonsense, and canonical splice site) were previously reported by our group [31]. We investigated 319 missense variants using bioinformatic criteria established for selecting variants of interest: uncommon (0.1–1% MAF) in non-cancer controls from gnomAD, highly conserved, and predicted to affect protein function by in silico tools. Three missense variants met these criteria (Figure 2): *FANCI* c.286G>A; p.E96K, c.1573A>G; p.M525V, and our candidate variant c.1813C>T; p.L605F. *FANCI* c.1573A>G; p.M525V was previously reported by our group; however, based on in cellulo assays, we concluded that this variant did not affect protein function [31]. Thus, the only *FANCI* variant to investigate further was c.286G>A; p.E96K. In the gnomAD non-cancer population, the allele frequency of c.286G>A is 0.17%, which is less common than the allele frequency of c.1813C>T of 0.67%. We investigated the available genetic data or genotyped FC, Australian, and TCGA study groups for carriers of *FANCI* c.286G>A; p.E96K. *FANCI* c.286G>A was not identified in OC (n = 527) or BC (n = 220) cases or controls (n = 171) of FC ancestry, but was previously reported by our group in an Australian OC case (1/516, 0.2%), controls (5/4878, 0.1%), and TCGA OC cases (1/412, 0.2%) [31].

### 3.6. Loss of Heterozygosity Analyses of FANCI c.1813C>T in OC Tumour DNA from Carriers

We previously reported loss of the wild-type allele in bilateral OC tumours from *FANCI* c.1813C>T carriers, suggesting that loss of FANCI function was an early event in tumourigenesis [31]. We extended our analysis to investigate tumour samples from other carriers, although only DNA from FC cases was available for these analyses. The inspection of Sanger sequencing chromatograms from OC tumour DNA from nine carriers revealed three cases exhibiting loss of the wild-type allele and retention of the *FANCI* c.1813C>T allele. One case showed loss of the variant allele and retention of the wild-type allele, and the remaining cases retained heterozygosity with little evidence of allelic imbalance.

### 3.7. Somatic Genetic Analyses of OC Tumours from FANCI c.1813C>T Carriers

The somatic genetic landscape of HGSC cases has been well characterized, where there is a long tail of uncommon somatic variants and extensive genome-wide CNAs, with the exception of *TP53* (which harbours driver mutations that cause cells to become cancerous) being the most altered gene (>95% of cases) [74]. To determine if HGSC cases carrying *FANCI* c.1813C>T exhibit similar somatic genetic characteristics to HGSC cases, we performed WES analyses or surveyed the available genetic data from seven FC cases and six TCGA cases, respectively. We focused our analyses on the most altered genes reported for HGSCs, *TP53* (96%), *BRCA1* (3.5%), *CSMD3* (6%), *NF1* (4%), *CDK12* (3%), *FAT3* (6%), *GABRA6* (2%), *BRCA2* (3%), and *RB1* (2%) [74]. Somatic variants were identified in *CDK12* (1/13, 8%), *FAT3* (3/13, 23%), *BRCA2* (3/13, 23%), and *TP53* (11/13, 85%) (Table 4), at frequencies comparable to those HGSC cases [74]. As expected, most deleterious variants identified in our HGSC cases carrying *FANCI* c.1813C>T in *TP53* were missense [74,107,108] (n = 7), with the remainder being frameshift (n = 2), splice (n = 1), or inframe indel (n = 1). Extensive and unremarkable genome-wide CNAs were evident across tumours from *FANCI* c.1813C>T carriers, which was consistent with those seen for HGSC tumours (Appendix A). Amplification of *CCNE1*, reported to occur in approximately 20% of HGSC cases [74], was exhibited in two *FANCI* carrier cases (2/11, 18%), PT0006 from our FC study group and sample TCGA-25-2393 from the TGCA project.

DNA from HGSC tumours has been shown to exhibit global DNA mutational signatures, reflecting disruptions in specific DNA repair pathways, aging, and other processes that have accumulated during tumourigenesis [79]. We performed a somatic mutational signature analysis using WES data derived from FC OC tumour DNA from *FANCI* c.1813C>T carriers, using COSMIC SBS signatures as a reference. The signatures identified in tumours from *FANCI* c.1813C>T carriers were compared to those exhibited by HGSC tumours, as there have been no reports attributing mutational signatures to cancers harbouring deleterious *FANCI* variants. We were able to profile seven OC tumour samples from FC carriers due to the availability of WES data for these samples (Appendix A). The mutational profiles were indicative of the presence of extensive and complex mutational patterns typified by HGSC tumours (https://signal.mutationalsignatures.com/explore/tissueType/15: accessed on 16 May 2022). The homologous recombination deficiency signature (referred to as SBS3) was identified in 6/7 (86%) cases. The sample that did not exhibit this SBS3 signature, PT0003, exhibited signature pattern SBS8, a signature whose etiology is unknown, but it has been proposed to be associated with homologous recombination deficiency (cancer.sanger.ac.uk/signatures/sbs/sbs8/: accessed on 16 May 2022). PT0003 also exhibited the largest contribution of signature SBS6, which has been attributed to defective mismatch repair and MSI. The aging signature (SBS1) was identified in 5 out of 7 (71%) tumours and the contribution was consistent with age at diagnosis [31]. All tumours exhibited varying proportions of SBS18, a signature indicative of damage due to reactive oxygen species. A signature with a proposed etiology associated with prior treatment with platinum chemotherapy drugs (SBS35) was evident in 5 out of 7 (71%) tumours, although not in sample PT0007, which was from a patient who received neoadjuvant chemotherapy with the platinum compound carboplatin.

### 3.8. Germline FANCI c.1813C>T Carriers Identified in Other Cancer Types

We previously reported carriers of *FANCI* c.1813C>T and other potentially pathogenic *FANCI* variants in BC cases, a disease associated with OC risk genes [109], and a review of the literature also indicated that there were *FANCI* carriers in other cancer types [31]. To further investigate the role of *FANCI* in other cancer types we investigated the germline carrier frequency of *FANCI* c.1813C>T in 10,389 cancer cases from the TCGA PanCancer data set [45]. We focused on this variant to further investigate its association with familial OC and our in cellulo assays, demonstrating abrogated protein function [31]. The highest carrier frequency was observed in adrenocortical carcinoma cases (3.3%, 3/92); there were no carriers identified in cases with diffuse large B-cell carcinoma (n = 41), kidney renal papillary cell carcinoma (n = 289), thymoma (n = 123), or uterine carcinosarcoma (n = 57) (Table 5). The median age of diagnosis (59 ± 14.7 years) and the number of females (48.5%) of *FANCI* c.1813C>T carriers were comparable to the total study group (59.2 ± 14.4 [45] and 52% [45], respectively). Interestingly, the overall carrier frequency of *FANCI* c.1813C>T at 1.6% (171/10,389) was significantly higher in the TCGA PanCancer cases than in non-cancer individuals in gnomAD (1.3%, 1787/134,164; Pearson’s χ^2^ = 7.3, *p* = 0.007).

### 3.9. A Wide Spectrum of Somatic FANCI Variants Identified in a Variety of Cancer Types

Approximately 40% of germline CPGs have been found with somatic variants in tumours as drivers and some of these genes have been shown to play a role in tumourigenesis, with *RB1* being the classical example [110]. From the above analysis of the germline *FANCI* variant, there were four *FANCI* c.1813C>T germline carriers with different somatic *FANCI* variants, two bladder urothelial carcinomas and two lung squamous cell carcinomas. These observations prompted us to investigate the spectrum and frequency of somatic variants in *FANCI* in TCGA PanCancer tumours (n = 10,434) from cBioPortal [46,47]. We identified 198 different variants in 172 tumours (1.65%, 172/10,434) in 28 different cancer types, comprising a variety of genetic abnormalities: 168 missense, 11 nonsense, 10 splice, 6 frameshift, 2 stop loss, and 1 inframe deletion variant (Table 6 and Appendix A). Noteworthy, *FANCI* c.1813C>T was not among the variants identified at the somatic level. There appears to be no mutational hotspot identified and variants were distributed across the gene (Figure 3). The highest total number of variants was identified in uterine corpus endometrial carcinoma tumours (8.32%, 43/517); no variants were identified in tumours from cases with cholangiocarcinoma (n = 36), diffuse large B-cell carcinoma (n = 41), kidney chromophobe (n = 65), pancreatic adenocarcinoma (n = 179), or uveal melanoma (n = 80). The median age at diagnosis and sex of cases with somatic *FANCI* variants were comparable to those without somatic *FANCI* variants (60 ± 13.5 vs. 60 ± 14.4 and 58% vs. 52%, respectively). Tumours with somatic *FANCI* variants had a higher mutational load (*p* = 2.2 × 10^−16^) and MSI score (*p* = 6.8 × 10^−10^) compared to tumours without somatic *FANCI* variants.

## 4. Discussion

Genetic analyses of the germline of two sisters with OC from family F1528 that are heterozygous for *FANCI* c.1813C>T; p.L605F revealed 222 variants of interest, of which 66 were most likely to exert a function on the encoded protein (genetic landscape variants). Of note is the number of loss-of-function (frameshift, nonsense, and canonical splice site) variants identified in both sisters after filtering and prioritization (n = 15). Increasing evidence shows that all individuals carry more potentially deleterious variants than previously suspected (approximately 24–100 heterozygous variants per individual [111,112]), although some may not impact gene function [113,114]. As adequate in silico tools with high predictive performance, such as those for missense and splice site variants, have not yet been developed for loss-of-function variants, it is difficult to further interpret these variants in the absence of laboratory experiments by examining their biological effect. The investigation of the genetic landscape variants in OC cases of FC ancestry revealed four carriers of variants in *PTPN22, GPD1,* and *SEC14L4* in three different families, although two of these families were previously identified by our group to harbour likely pathogenic variants in known or putative DNA repair pathway genes (P.N. Tonin unpublished data). When we assessed the same study group for carrier status, we identified six carriers of variants in nine different genes, where all but two families were previously identified to harbour likely pathogenic variants in DNA repair genes (P.N. Tonin unpublished data). Families F845 and F1543 were the only cases not found to harbour likely pathogenic variants in DNA repair genes. Although the previously identified DNA repair pathway variants are plausible OC-predisposing variants, they have yet to be verified independently. Although it is possible that the genetic variants identified in this study may independently affect the risk of OC, *FANCI* c.1813C>T; p.L605F remains the most likely candidate for an OC-predisposing gene in family F1528.

We also investigated the 66 genetic landscape variants for carrier frequency in the Australian OC cases (regardless of *FANCI* variant carrier status) and identified several carriers of variants in these genes, mainly missense variants. These variants had not been reported in a previous independent analysis of WES data from these cases, as the study focused primarily on loss-of-function variants [44]. Thus, it is possible that these missense variants are relevant in these cases, which are notably negative for pathogenic variants in *BRCA1* and *BRCA2*. Also notable is that we did not identify any carriers of any of our genetic landscape variants among the 10 previously identified *FANCI* c.1813C>T; p.L605F carriers from this study group. Based on principal component analysis (PCA), the Australian OC cases are likely of white European ancestry, which is similar to the ancestral origins of the FC population [31].

The role of the 66 genetic variants identified in the *FANCI* carriers in modifying the risk of OC in family F1528, which harbours *FANCI* c.1813C>T; p.L605F, remains to be determined. Interestingly, additional potentially pathogenic variants were identified in *FANCI* using the criteria established for c.1813C>T and it would be of interest to further assess their effect on gene function. The most interesting variant, *FANCI* c.286G>A; p.E96K, was identified in a BC case (diagnosed at 44 and 50 years), with OC at age 53 years, was of Ukrainian ancestry, and who reported a grandmother with OC (Appendix A). This individual did not harbour any pathogenic variants in *BRCA1*, *BRCA2*, *BRIP1*, *RAD51C*, or *RAD51D* by WES analyses. *FANCI* c.286G>A; p.E96K was reported in BC cases (2/133, 1.5%) [115] and OC cases (1/6385, 0.02% (0/6115 controls)) [116] (p. 2) in the literature. FANCI p.E96K may affect the ubiquitination of FANCD2, the heterodimeric binding partner of FANCI, and/or the Van der Waals forces between FANCI and FANCD2 [115].

HGSC tumours from cases harbouring germline *FANCI* c.1813C>T exhibited features consistent with tumours from HGSC cases. The majority (85%) of tumours had identifiable pathogenic variants in *TP53* and two cases were identified with *CCNE1* amplification (18%). The somatic mutational signatures characteristic of HGSC tumour cells were also present in HGSC samples harbouring *FANCI* c.1813C>T. We also identified signatures SBS35 (platinum associated) and SBS6 (mismatch repair deficiency) in 5 out of 7 and 6 out of 7 cases, respectively, which are less commonly observed in HGSC cases (<10%). This may be the result of the small sample size, although they could also be the result of harbouring *FANCI* c.1813C>T in the germline. Interestingly, the two cases with the highest contributions of the SBS6 signature, samples PT0004 and PT0005, have germline or somatic variants in mismatch repair genes, which may contribute to the presence of this signature. In a study involving *Caenorhabditis elegans,* it has recently been reported that genotoxic agents tended to have a stronger influence on the mutational signature than a DNA repair-deficient background [117]. Moreover, the signature attributed to defects in HR DNA repair may be identified in the absence of an identifiable DNA repair pathway gene pathogenic variant [118]. This could be because the signature is more attributed to global defects and is distinct from other signatures that have characteristic nucleotide changes such as the aging signature [118]. Although the sample size was small, there appeared to be no identifiable *FANCI*-specific signature from the analyses of tumours from *FANCI* c.1813C>T carriers.

The contribution of cancer cases attributed to CPGs is approximately 3%, although this varies based on cancer type [110]. Some CPGs predispose to multiple primary cancer types, such as *BRCA1* with OC and BC, although there is often a preferential predisposition to certain histological subtypes such as the association of *BRCA1* with HGSC. Germline *FANCI* c.1813C>T was initially identified in HGSC cases [31], but as shown in our study, it can be observed across many cancer types in the TGCA PanCancer Atlas. It is unknown if *FANCI* c.1813C>T contributes to the risk of these cancers, as the variant is more common in the general population compared to other high-risk CPGs (0.6% vs. 0.0001%) [48]. The moderately increased carrier frequency of *FANCI* c.1813C>T in TCGA cancer cases (1.6%) compared to gnomAD cancer-free controls (1.3%) is intriguing, suggesting a role for this variant in risk (*p* = 0.007). Additionally, TCGA cancer cases were identified with somatic variants in *FANCI* across cancer types. As variants were identified in cases with significantly higher mutational load and/or MSI scores, it is possible that these somatic *FANCI* variants arose as a consequence of either of these processes, although they could be drivers of these processes through mechanisms that remain to be elucidated. *FANCI* is not currently included among the 733 cancer-driving genes in the Cancer Gene Census [119], but a number of other genes involved in the FA pathway, such as *BRCA1 (FANCS)*, *BRCA2 (FANCD1)*, *BRIP1 (FANCJ)*, *FANCA*, *FANCC*, *FANCD2*, *PALB2 (FANCN),* and *RAD51C (FANCO)*, are implicated as cancer drivers in this census. Further assessment of cancers in individuals harbouring germline *FANCI* c.1813C>T or somatic *FANCI* variants may elucidate a genetic signature associated with *FANCI*, as has been reported for FA-associated squamous cell carcinomas [120].

Many CPGs are also associated with non-cancer phenotypes, the spectrum of which is broad. Homozygous or compound heterozygous variants in *FANCI* were associated with FA complementation group I in 2007 [99,100,121]. FA is a rare disease characterized by congenital defects, progressive bone marrow failure, and an increased risk of cancers (mainly acute myeloid leukemia and squamous cell carcinomas) [122]. FA-I cases comprise approximately 1% of all FA cases and have been associated with at least three features of VACTERL-H [123], a rare disease that affects multiple body systems. Recently, an eight-year-old male with aplasia referred for a diagnosis of FA was reported to harbour germline homozygous *FANCI* c.1813C>T [124]. FANCD2 ubiquitination was not detected in peripheral blood cells from this patient and increased chromosomal breakage was observed, suggesting abrogation of the FA pathway. These data are consistent with our previous observations that *FANCI* c.1813C>T abrogates FANCI protein function [31]. FANCI is an integral member of the FA pathway and acts as the molecular switch to activate the pathway [101]. FANCI also functions outside the FA pathway such as in dormant origin firing [125], negative regulation of Akt signalling [126], and ribosome biogenesis [127].

The risk of cancer has been assessed in heterozygous relatives of individuals with FA, and although no association with cancer risk was found, few *FANCI* families (n = 4) have been investigated due to the paucity of *FANCI* carriers [128,129,130]. *FANCI* c.1813C>T; p.L605F was previously identified by our group in the germline and was associated with a suspected autosomal dominant mode of inheritance of OC [31]. This is consistent with more than half of CPGs, which are associated with an autosomal dominant mode of inheritance [110]. The majority of CPGs also act as tumour suppressors, where many are classical tumour suppressors that require biallelic inactivation for tumour development and/or progression, although some CPGs may exert their effect through haploinsufficiency or in a dominant-negative manner [110]. Here, we have shown that biallelic inactivation of *FANCI* c.1813C>T may occur through loss of the wild-type allele. As we showed that not all carriers exhibited this loss, it is possible that loss of the wild-type allele may not be required for OC tumourigenesis. This is consistent with the OC-predisposing genes *BRCA1* and *BRCA2*, where loss of the wild-type allele is not always observed in tumour cells from carriers of pathogenic variants in these CPGs [22,131]. Beyond assessing the biological effect of a variant using cell-line models, as we have shown with *FANCI* c.1813C>T [31], there are no suitable animal models to evaluate OC risk alleles.

The identification of CPGs has had a significant clinical impact on diagnosis and management, targeted therapies, and screening and prevention. The clinical utility of *FANCI* for diagnosis and management cannot be determined until penetrance for cancer risk is established. Currently, there are no effective cancer screening methods for OC or prevention strategies to reduce OC risk in the general population, although prophylactic salpingo-oophorectomy has been proven to reduce risk in carriers of pathogenic *BRCA1* or *BRCA2* variants [132]. Given the identification of FANCI variants in other cancers, particularly gynecological cancers, future research should investigate the risks of these variants in view of prevention and management strategies, including childbearing considerations [133,134]. Although there are no targeted therapies for *FANCI*, 73 chemicals interact with this gene, including cisplatin and mitomycin C, which is concordant with our previous findings that loss of *FANCI* sensitizes cells to these drugs [31]. There are 12 cancer-related drugs that interact with FANCI, 7 of which are chemotherapies and 5 of which are targeted therapies [135]. These chemicals present opportunities for future investigation for the treatment of cancer cases with *FANCI* variants.

## 5. Conclusions

This study has expanded on the molecular genetic characteristics of *FANCI* c.1813C>T; p.L605F in OC, which were first reported in OC families of FC ancestry [31]. These data suggest *FANCI* c.1813C>T carrier HGSC tumours show characteristics known to be exhibited by HGSC cases. The identification of germline *FANCI* c.1813C>T carriers and various somatic *FANCI* variants across cancer types suggests a possible involvement of *FANCI* in other cancers and an avenue for future research.

## Figures and Tables

**Figure 1 genes-14-00277-f001:**
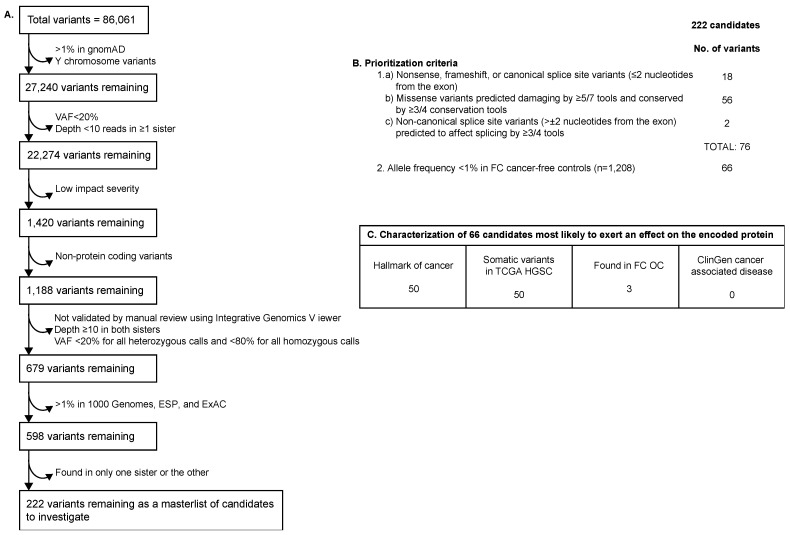
Criteria used for filtering and prioritizing variants identified across the genetic landscape of sisters from family F1528. (**A**) Filtering strategy to identify genetic landscape variants; (**B**) prioritization of variants to identify those most likely to exert an effect on the encoded protein; and (**C**) characterization of variants most likely to affect protein function using various characteristics of cancer-associated genes. gnomAD: Genome Aggregation Database; VAF: variant allele frequency; ESP: Exome Sequencing Project; ExAC: Exome Aggregation Consortium; FC: French Canadian; TCGA: The Cancer Genome Atlas; HGSC: high-grade serous ovarian carcinoma; OC: ovarian cancer.

**Figure 2 genes-14-00277-f002:**
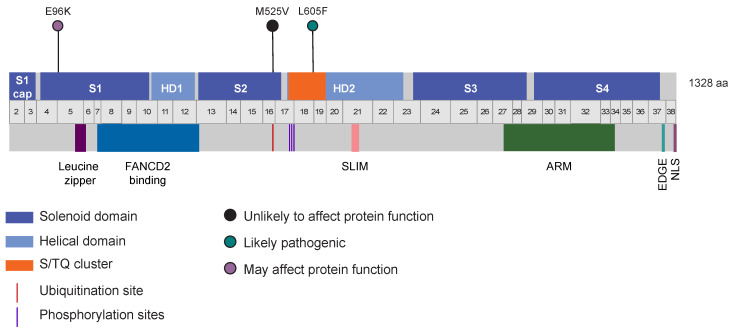
Schema showing location of variants in *FANCI* gene and protein identified from ClinVar using the established criteria for c.1813C>T; p.L605F. FANCI domains were adapted from pfam (https://pfam.xfam.org). *FANCI* exon locations were adapted from the University of California Santa Cruz Genome Browser (https://genome.ucsc.edu). Solenoid domain: antiparallel pairs of α-helices that form α-α superhelix segments; Helical domain: α-helices; Ubiquitination site: site of monoubiquitination by the FA core complex to allow downstream FA pathway function, located at K523 [99,100]; S/TQ cluster: location of conserved phosphorylation sites [101]. Phosphorylation sites (556, 559, and 565aa): sites of phosphorylation that stabilize the association of FANCI with DNA and FANCD2 [102]. Leucine zipper (130–151aa): may be related to protein–protein interactions, DNA binding, or RNA binding, but the leucine zipper found at the N-terminus of FANCI has been shown not to bind to DNA [103]. Ubiquitin binding (175–377aa): this region binds to the ubiquitin on FANCD2 [104]. SUMO-like domain-interacting motif (SLIM; 682–696aa): binds to the SUMO-like domain 2 (SLD2) of UAF1 promoting FANCD2 deubiquitination, which is required for FA pathway function [105]. Armadillo repeat (ARM; 985–1207aa): forms a superhelix of helices, which can also be found in FANCD2 [100]. EDGE motif (1300–1303aa): this motif consists of Glutamic acid (E)–Aspartic acid (D)–Glycine (G)–Glutamic acid (E) and is required for DNA crosslink repair function [99,100,106]. Nuclear localization site (NLS; 1323–1238aa): required for localization to the nucleus, where subsequent function in the FA pathway can occur [106].

**Figure 3 genes-14-00277-f003:**
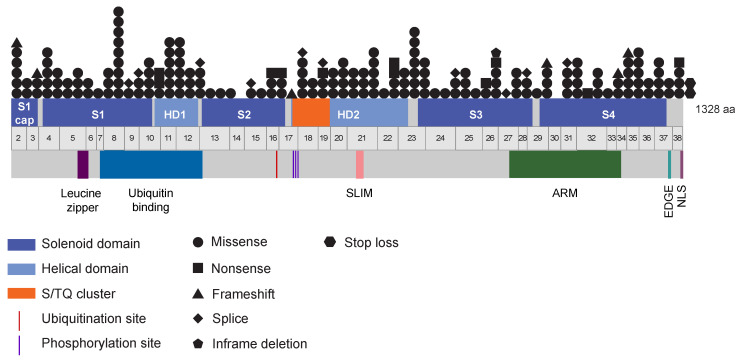
Schema showing the location of all somatic variants in the FANCI gene and protein identified in tumours from the TCGA PanCancer Atlas.

**Table 1 genes-14-00277-t001:** Genetic landscape variants and other variants identified in genes where genetic landscape variants were identified in OC cases of FC ancestry negative for pathogenic variants in *BRCA1*, *BRCA2*, *BRIP1*, *RAD51C*, and *RAD51D*, including *FANCI* c.1813C>T carrier status.

Gene	Coding Change	Protein Change	F1528-PT0056 ^1^	F1528-PT0057 ^1^	F1085-PT0134 ^2^	F1085-PT0135 ^2^	F1601-PT0138	F845-PT0196	F1490-PT0047	F1620-PT0100	F1506-PT0136	F1543-PT0137	F1288-PT0158	F1617-PT0090	F694-PT0128	F439-PT0184	F1650-PT0142
*FANCI*	c.1813C>T	p.L605F	x	x					x	x							
*PTPN22*	c.993-1G>A	NA	x	x	x	x											
*GPD1*	c.431T>C	p.M144T	x	x			x										
*SEC14L4*	c.364C>T	p.R122W	x	x				x									
*PIWIL3*	c.2023T>G	p.C675R	x	x													
*PIWIL3*	c.1932+1G>A	NA									x						
*CACNA1S*	c.4340G>A	p.R1447Q	x	x													
*CACNA1S*	c.773G>A	p.G258D			x												
*MYO7A*	c.5866G>A	p.V1956I	x	x													
*MYO7A*	c.1078G>T	p.E360 *			x												
*SCN10A*	c.3776G>A	p.R1259Q	x	x													
*SCN10A*	c.2972C>T	p.P991L											x				
*PCDH15*	c.3127C>T	p.P1043S	x	x													
*PCDH15*	c.2581G>A	p.V861M						x									
*TEX2*	c.73G>T	p.V25L	x	x													
*TEX2*	c.3040G>A	p.E1014K						x									
*DNAH3*	c.10382C>G	p.P3461R	x	x													
*DNAH3*	c.5368A>T	p.I1790F								x							
*DNAH1*	c.1941_1944del	p.N648Afs * 36	x	x													
*DNAH1*	c.2717A>G	p.D906G										x					
*DNAH1*	c.10216G>A	p.V3406I									x						
*IQCA1*	c.29G>A	p.W10 *	x	x													
*IQCA1*	c.979G>C	p.A327P										x					

x: heterozygous; NA: not applicable; ^1^ From the same family F1528; ^2^ From the same family F108. * Denotes a change to a stop codon.

**Table 2 genes-14-00277-t002:** Genetic landscape variants identified in Australian HGSC cases (n = 516).

Gene	Coding Change	Protein Change	No. of Carriers (%)
*CACNA1S*	c.4340G>A	p.R1447Q	1 (0.2)
*NBAS*	c.3217C>T	p.R1073C	6 (1.2)
*ANKAR*	c.3815G>A	p.R1272H	2 (0.4)
*PARD3B*	c.365T>C	p.I122T	1 (0.2)
*TNS1*	c.1333G>C	p.G445R	4 (0.8)
*IQCA1*	c.29G>A	p.W10 *	3 (0.6)
*CXCL6*	c.239dup	p.V81Gfs * 44	9 (1.7)
*CEP120*	c.2134C>T	p.L712F	8 (1.6)
*KCNU1*	c.2731G>A	p.A911T	1 (0.2)
*NUP188*	c.3974G>A	p.R1325H	4 (0.8)
*CREM*	c.677C>T	p.S226L	5 (1)
*PCDH15*	c.3127C>T	p.P1043S	1 (0.2)
*NPFFR1*	c.8-2A>G	NA	6 (1.2)
*MYO7A*	c.5866G>A	p.V1956I	3 (0.6)
*PWP1*	c.1402G>A	p.E468K	7 (1.4)
*PAQR5*	c.20C>G	p.P7R	3 (0.6)
*DNAH3*	c.10382C>G	p.P3461R	1 (0.2)
*PLIN4*	c.3260_3263dup	p.F1089Pfs * 32	3 (0.6)
*CYP2A6*	c.289G>A	p.E97K	1 (0.2)
*ALDH16A1*	c.1376A>T	p.D459V	5 (1)
*MYH9*	c.4396C>T	p.R1466W	2 (0.4)

* Denotes a change to a stop codon.

**Table 3 genes-14-00277-t003:** FANCI interactome candidate variants identified in FC OC cases negative for pathogenic variants in *BRCA1*, *BRCA2*, *BRIP1*, *RAD51C*, *RAD51D*, and *FANCI*.

Gene	Coding Change	Protein Change	F1601-PT0138	F1506-PT0136	F845-PT0196	F1085-PT0134 ^1^	F1085-PT0135 ^1^	F1617-PT0090	F694-PT0128	F1288-PT0158	F1543-PT0137	F439-PT0184	F1650-PT0142
*EZH2*	c.1786G>A	p.Ala596Thr	x										
*ANKRD55*	c.1126T>C	p.Ser376Pro		x									
*MOV10*	c.2501G>A	p.Arg834Gln		x									
*LRRK2*	c.356T>C	p.Leu119Pro			x								

^1^ From the same family F1085.

**Table 4 genes-14-00277-t004:** Somatic variants in the nine most frequently altered genes in HGSC identified in cases harbouring *FANCI* c.1813C>T (n = 13).

Sample ID	*TP53*	*BRCA1*	*CSMD3*	*NF1*	*CDK12*	*FAT3*	*GABRA6*	*BRCA2*	*RB1*
PT0001									
PT0002									
PT0003									
PT0004									
PT0006									
PT0005									
PT0007									
TCGA-04-1336									
TCGA-24-1603									
TCGA-25-2393									
TCGA-29-2431									
TCGA-61-1903									
TCGA-61-2009									
Total (%)	11 (85%)	0	0	0	1 (8%)	3 (23%)	0	3 (23%)	0

Colors legend: green means missense; yellow means splice; violet means frameshift; blue means in-frame.

**Table 5 genes-14-00277-t005:** Carrier frequency of *FANCI* c.1813C>T in TCGA PanCancer cases (n = 10,389).

Cancer Type (TCGA Acronym)	Total No. Cases	No. of *FANCI* c.1813C>T Carriers	Carrier Frequency of *FANCI* c.1813C>T (%)
Adrenocortical carcinoma (ACC)	92	3	3.3
Kidney chromophobe (KICH)	66	2	3
Lung squamous cell carcinoma (LUSC)	499	14	2.8
Skin cutaneous melanoma (SKCM)	470	13	2.8
Kidney renal clear cell carcinoma (KIRC)	387	10	2.6
Colon adenocarcinoma (COAD)	419	10	2.4
Cholangiocarcinoma (CHOL)	45	1	2.2
Esophageal carcinoma (ESCA)	184	4	2.2
Brain lower-grade glioma (LGG)	515	11	2.1
Liver hepatocellular carcinoma (LIHC)	375	8	2.1
Head and neck squamous cell carcinoma (HNSC)	526	9	1.7
Uterine corpus endometrial carcinoma (UCEC)	543	9	1.7
Breast invasive carcinoma (BRCA) ^1^	1076	17	1.6
Cervical squamous cell carcinoma (CESC)	305	5	1.6
Sarcoma (SARC)	255	4	1.6
Stomach adenocarcinoma (STAD)	443	7	1.6
Ovarian serous cystadenocarcinoma (OV)	412	6	1.5
Testicular germ cell tumours (TGCT)	134	2	1.5
Lung adenocarcinoma (LUAD)	518	7	1.4
Rectum adenocarcinoma (READ)	145	2	1.4
Uveal melanoma (UVM)	80	1	1.3
Bladder urothelial carcinoma (BLCA)	412	5	1.2
Mesothelioma (MESO)	82	1	1.2
Thyroid carcinoma (THCA)	499	6	1.2
Pancreatic adenocarcinoma (PAAD)	185	2	1.1
Pheochromocytoma and paraganglioma (PCPG)	179	2	1.1
Glioblastoma multiforme (GBM)	393	4	1
Prostate adenocarcinoma (PRAD)	498	5	1
Acute myeloid leukemia (LAML)	142	1	0.7
Diffuse large B-cell carcinoma (DLBC)	41	0	0
Kidney renal papillary cell carcinoma (KIRP)	289	0	0
Thymoma (THYM)	123	0	0
Uterine carcinosarcoma (UCS)	57	0	0
Total	10,389	171	1.6
gnomAD non-cancer overall ^2^	134,164	1787	1.3

^1^ 1 homozygous carrier; ^2^ 17 homozygous carriers.

**Table 6 genes-14-00277-t006:** Frequency of somatic *FANCI* variants identified in TCGA PanCancer tumours.

Cancer type (TCGA Acronym)	Total No. of Cases	No. of Tumours Harbouring *FANCI* Variants	Frequency of Tumours Harbouring *FANCI* Variants (%)
Uterine corpus endometrial carcinoma (UCEC)	517	43	8.32
Skin cutaneous melanoma (SKCM)	438	20	4.57
Bladder urothelial carcinoma (BLCA)	410	13	3.17
Colon adenocarcinoma (COAD)/Rectum adenocarcinoma (READ)	534	14	2.62
Stomach adenocarcinoma (STAD)	436	9	2.06
Cervical squamous cell carcinoma (CESC)	291	6	2.06
Uterine carcinosarcoma (UCS)	57	1	1.75
Lung squamous cell carcinoma (LUSC)	484	8	1.65
Lung adenocarcinoma (LUAD)	566	9	1.59
Head and neck squamous cell carcinoma (HNSC)	515	8	1.55
Acute myeloid leukemia (LAML)	200	3	1.50
Mesothelioma (MESO)	86	1	1.16
Adrenocortical carcinoma (ACC)	91	1	1.10
Esophageal carcinoma (ESCA)	182	2	1.10
Glioblastoma multiforme (GBM)	391	4	1.02
Breast invasive carcinoma (BRCA)	1066	10	0.94
Thymoma (THYM)	123	1	0.81
Sarcoma (SARC)	255	2	0.78
Ovarian serous cystadenocarcinoma (OV)	523	4	0.76
Kidney renal clear cell carcinoma (KIRC)	402	3	0.75
Testicular germ cell tumours (TGCT)	149	1	0.67
Pheochromocytoma and paraganglioma (PCPG)	178	1	0.56
Liver hepatocellular carcinoma (LIHC)	366	2	0.55
Prostate adenocarcinoma (PRAD)	494	2	0.40
Brain lower-grade glioma (LGG)	514	2	0.39
Kidney renal papillary cell carcinoma (KIRP)	276	1	0.36
Thyroid carcinoma (THCA)	489	1	0.20
Cholangiocarcinoma (CHOL)	36	0	0
Diffuse large B-cell carcinoma (DLBC)	41	0	0
Kidney chromophobe (KICH)	65	0	0
Pancreatic adenocarcinoma (PAAD)	179	0	0
Uveal melanoma (UVM)	80	0	0
Total	10,434	172	1.65

## Data Availability

The data sets generated and/or analyzed during the current study are not publicly available due to ongoing analyses as part of current projects but are available from the corresponding author on reasonable request.

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
