# Peer review of "Molecular Genetic Characteristics of FANCI, a Proposed New Ovarian Cancer Predisposing Gene"

_genes, 2023, doi:10.3390/genes14020277_

Round 1

Reviewer 1 Report

This manuscript is interesting in terms of investigating the molecular genetic characteristics of FANCI which might be the candidate of one of the cancer-predisposing genes although the clinical utility of FANCI for diagnosis and management was undetermined. For a better understanding of this manuscript, several revisions will be helpful.

Do Table 1 and 2 need to separate? Why were they separated?

In Results 3.5, the authors showed the frequency of FANCI c, 286G>A; p.E96K. It is more interesting if the authors compared it with the variant c.1813C>T; p.L605F.

In Results 3.7, how is the frequency of somatic FANCI c.1813C>T variant in HGSC?

In Discussion, page 16 lines 23-25, the authors insisted that FANCI c.1813C>T; p.L605F remained the most likely candidate OC predisposing gene in family F1528, however; it is difficult to understand without the previous manuscript. Further explanation is helpful for better understanding.

In Discussion, page 16 lines, the authors did not identify any carriers of any of their genetic landscape variants among the 10 previously identified FANCI c.1813C>T; p.L605F carriers from this group. Is the ethnic difference considered?

In Discussion, page 17 line 39, what is FA group I?

How about the perspective of FANCI as the candidate of OC predisposing genes? Show the authors’ opinions concisely.

Author Response

Please see our responses below in red

This manuscript is interesting in terms of investigating the molecular genetic characteristics of FANCI which might be the candidate of one of the cancer-predisposing genes although the clinical utility of FANCI for diagnosis and management was undetermined. For a better understanding of this manuscript, several revisions will be helpful.

Do Table 1 and 2 need to separate? Why were they separated?

Though they address different questions, we agree with the merits of that combining the data in tables 1 and 2 would allow for cataloging of all the variants identified in our study. The tables have now been combined. The table legend has been revised accordingly and all subsequent tables re-labelled.

In Results 3.5, the authors showed the frequency of FANCI c, 286G>A; p.E96K. It is more interesting if the authors compared it with the variant c.1813C>T; p.L605F. 

We agree, and for clarity we added the below sentence to section 3.5

“In the gnomAD non-cancer population, the allele frequency of c.286G>A is 0.17%, which is less common than the allele frequency of c.1813C>T at 0.67%.”

In Results 3.7, how is the frequency of somatic FANCI c.1813C>T variant in HGSC?

In results section 3.7, we discussed the somatic mutational spectrum only in carriers of germline FANCI c.1813C>T.

However, we discussed the frequency of all somatic variants identified in TCGA database in section 3.9, which would include c.1813C>T if reported. This variant was not identified somatically in this database. We added the below sentence to clarify this in section 3.9:

“Noteworthy, FANCI c.1813C>T was not among the variants identified at the somatic level.”

In Discussion, page 16 lines 23-25, the authors insisted that FANCI c.1813C>T; p.L605F remained the most likely candidate OC predisposing gene in family F1528, however; it is difficult to understand without the previous manuscript. Further explanation is helpful for better understanding.

The previous manuscript clearly details the identification of this variant and includes functional analyses, which combined support FANCIc.1813C>T’s potential as a candidate OC predisposing gene. The previously published paper (Fierheller et al. Genome Medicine 2021 PMID: 34861889) is quite substantive and has a robust discussion on the candidacy of this variant as an OC predisposing gene according to data presented in prior study.

This information is also summarized in the introduction to this manuscript under review. The last sentence in the first paragraph of the discussion is phrased in a way to leave open the possibility that there may be other variants that could be contributing to OC risk in carriers, as would be the case for carriers of any of the known OC predisposing genes. This notion is raised but not discussed in the prior paper and is the subject of the current study.

In Discussion, page 16 lines, the authors did not identify any carriers of any of their genetic landscape variants among the 10 previously identified FANCI c.1813C>T; p.L605F carriers from this group. Is the ethnic difference considered?

The ethnicity wasn’t specifically considered in this analysis, however, we noted the ancestral origins of both the Australians and FCs in the discussion:

“Based on principal component analysis (PCA), the Australian OC cases are likely of white European ancestry which is similar to the ancestral origins of the FC population [31].”

In Discussion, page 17 line 39, what is FA group I?

This has been changed to FA complementation group I.

How about the perspective of FANCI as the candidate of OC predisposing genes? Show the authors’ opinions concisely.

As mentioned above, we provided an argument for FANCI as an OC predisposing gene in our previous publication, which described the discovery of FANCI c.1813C>T. As this has already been discussed, the purpose of this manuscript was to follow through with the molecular and genetic analyses of c.1813C>T and FANCI. As mentioned in this study, the original genetic data and functional work published previously combined with further genetic and molecular work presented and discussed in this study also support the candidacy of FANCI as an OC predisposing gene.

Reviewer 2 Report

Hello

Thank you to the editor for giving me the opportunity to review this interesting manuscript regarding the genetic risk of ovarian cancer. 

The text appears well written and the English appears appropriate. 

Given the importance of the topic, I think it is useful to add in the conclusions how important genetic evaluation is not only in ovarian cancer but also in other gynecological cancers and how the formation of risk categories based on genetic screening can personalize treatment (see Endometrial Cancer Experience; "Fertility Sparing Treatments in Endometrial Cancer Patients: The Potential Role of the New Molecular Classification." International journal of molecular sciences vol. 22,22 12248. 12 Nov. 2021, doi:10.3390/ijms222212248). 

In addition, I think it is important to add how genetic evaluation and screening for high-risk patients can allow identification of patients at risk for rare malignancies and allow specific treatment for such patients, especially if they are still of childbearing age (see "Adult Granulosa Cell Tumor in Pregnancy: A New Case and a Review of the Literature." Healthcare (Basel, Switzerland) vol. 9,11 1455. Oct. 27, 2021, doi:10.3390/healthcare9111455). 

Author Response

Please see our response below in red.

Hello

Thank you to the editor for giving me the opportunity to review this interesting manuscript regarding the genetic risk of ovarian cancer. 

The text appears well written and the English appears appropriate. 

Given the importance of the topic, I think it is useful to add in the conclusions how important genetic evaluation is not only in ovarian cancer but also in other gynecological cancers and how the formation of risk categories based on genetic screening can personalize treatment (see Endometrial Cancer Experience; "Fertility Sparing Treatments in Endometrial Cancer Patients: The Potential Role of the New Molecular Classification." International journal of molecular sciences vol. 22,22 12248. 12 Nov. 2021, doi:10.3390/ijms222212248). 

In addition, I think it is important to add how genetic evaluation and screening for high-risk patients can allow identification of patients at risk for rare malignancies and allow specific treatment for such patients, especially if they are still of childbearing age (see "Adult Granulosa Cell Tumor in Pregnancy: A New Case and a Review of the Literature." Healthcare (Basel, Switzerland) vol. 9,11 1455. Oct. 27, 2021, doi:10.3390/healthcare9111455). 

This topic is beyond the scope of our manuscript as we have not yet assessed the penetrance of this variant in OC or other gynecologic cancer where this variant has been determined, as mentioned in the discussion section. However, to stress the clinical significance for management of carriers we have included the following sentence in the last paragraph of the discussion section, which include the two references provided:

“Given the identification of FANCI variants in other cancers, particularly gynecological cancers, future research should investigate the risk of these variants in view of prevention and management strategies, including childbearing considerations[137], [138].”

Round 2

Reviewer 1 Report

The authors revised the manuscript according to the reviewer's comments.